# Real-Time Simulation of Deformable Tactile Sensors in Robotic Grasping using Graph Neural Networks

Guillaume Duret[1,3], Danylo Mazurak[1], Frederik Heller[3], Florence Zara[2], Jan Peters[3] and Liming Chen[1]

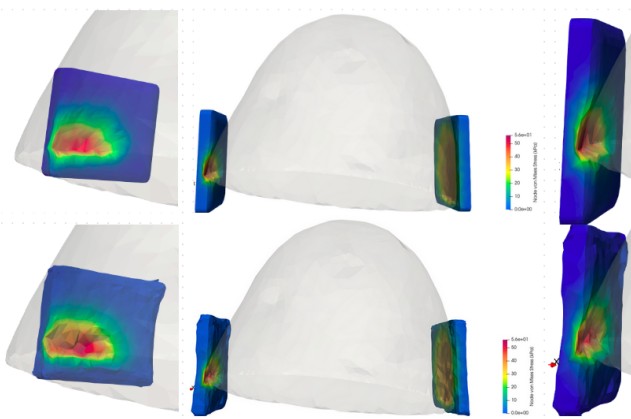

Fig. 1: Illustration of the ground truth stress and deformation of the tactile sensor across different grasping poses. The first row shows the ground truth, while the second row presents the prediction from the Graph Neural Network.

## I. INTRODUCTION

Physical simulation plays a crucial role in the development of robotic manipulation methods, and its importance is even greater when dealing with visual tactile sensors in contact-rich scenarios. Despite years of research, simulating such sensors remains highly challenging—both in terms of the underlying physical dynamics and the rendering of tactile images. In this work, we focus exclusively on the *physical simulation* aspect, leaving the rendering problem outside the scope of our study.

Related work on visual tactile sensor simulation can be broadly divided into two categories: (i) *rigid-body* simulations and (ii) *soft-body* simulations. Rigid-body simulations prioritize execution speed, making them suitable for scenarios requiring large-scale data generation, such as reinforcement learning [6], [1]. In contrast, soft-body approaches offer greater realism by capturing shear forces and deformations under contact with external objects [8]. However, they are significantly more computationally expensive and orders of

magnitude slower than rigid-body simulations.

This work addresses the speed limitations of soft-body simulations by leveraging Graph Neural Networks (GNNs) [4]. We explore the use of GNN models for simulating grasping interactions with visual tactile sensors, achieving performance gains between $10^2$ and $10^3$ times faster than traditional FEM simulations, while predicting both deformation and stress on the sensor. The code is available at: https://tacgraspnets.github.io

The main contributions of this paper are:

- The first application of GNN-based physics learning to visual tactile sensor simulation.
- A prediction framework capable of accelerating grasping simulations by a factor of $10^2$–$10^3$ compared to FEM, while generalizing to unseen grasping poses.

## II. DATASET GENERATION

For dataset generation (Fig. 2), we employ a Finite Element Method (FEM) simulation in Isaac Gym [7], building on Defgraspsim [5] but extending it to dynamic grasping scenarios using *TacGraspSim* [2]. This framework simulates parallel grippers with GelSight Mini tactile sensors with URDF format.

The simulation procedure is: (1) load object, (2) load gripper at grasping position, (3) close gripper until contact and (4) gradually increase grasp force to threshold $N$ while recording data. We save 50 frames per run, capturing: (i) node-wise deformations, (ii) rigid body poses, (iii) stress distributions and (iv) finger gripper translations for training.

## III. GRAPH NEURAL NETWORK METHOD (GNN)

In this work, we employ a Graph Neural Network (GNN) as the central component for learning the interactions between a deformable gripper and a rigid object. This choice is motivated by the demonstrated performance of GNNs in learning physics-based simulations, as seen in works like DefGraspNet [4] and the improved baseline used in this study [3]. GNNs are particularly suitable as they naturally model a mesh as a graph. Furthermore, their internal message-passing procedure propagates information through the graph in a manner analogous to Finite Element Method (FEM) simulations.

The graph structure contains nodes for both gripper (tetrahedral mesh vertices) and object geometry. Two edge types capture different interactions: *mesh edges* connect neighboring nodes within each structure, while *contact edges* connect object-gripper nodes to model contact (Fig. 2).

*This work was in part supported by the French Research Agency, l'Agence Nationale de Recherche (ANR), through the projects Learn Real (ANR-18-CHR3-0002-01), Chiron (ANR-20-IADJ-0001-01), Aristotle (ANR-21-FAI1-0009-01), and Astérix (ANR-23-EDIA-0002). It was granted access to the HPC resources of IDRIS under the allocation : 2025-[AD011015271R1], 2025-[AD011015591R1] made by GENCI.

[1]Centrale Lyon, CNRS, LIRIS, UMR5205, F-69130 Ecully, France, `guillaume.duret@ec-lyon.fr`

[2]UCBL, CNRS, LIRIS, UMR5205, F-69622 Villeurbanne, France

[3]Intelligent Autonomous Systems Lab, Technical University of Darmstadt, 64289 Darmstadt, Germany

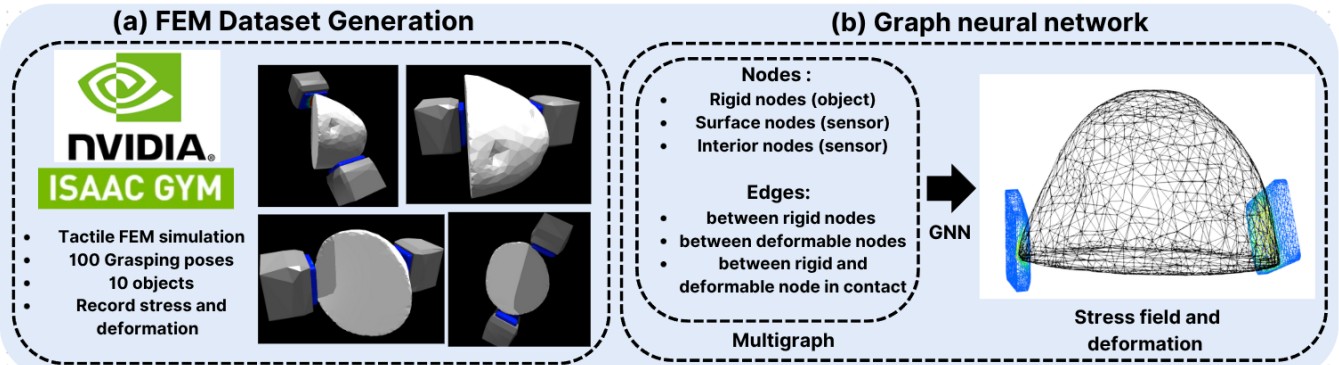

Fig. 2: Pipeline illustrating: (a) dataset creation using FEM simulation [7], [5], [2] across 10 different objects and 100 grasping poses per object, and (b) construction of the Graph Neural Network with node and edge descriptions to predict stress and deformation outputs.

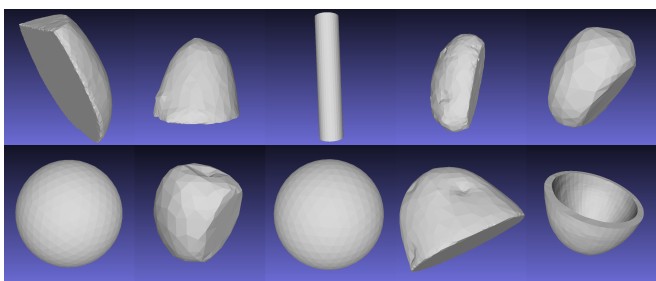

Fig. 3: Visualization of the 10 objects used in the dataset

| Class | Trans | Mean Def MAE | Mean Stress MAE |
|-------|-------|--------------|-----------------|
| potato | True | **6.57e-05** | **372.7** |
| potato | False | 2.92e-04 | 382.8 |
| apple | True | **7.20e-05** | **370.5** |
| apple | False | 2.97e-04 | 427.9 |
| lemon | True | **5.40e-05** | **212.1** |
| lemon | False | 2.38e-04 | 265.6 |

TABLE I: Single-object training results comparing models with and without translation input. Bold indicates superior performance.

Following an Encode-Process-Decode scheme, node features encode geometric state, type, and motion cues. Edge features contain geometric information (mesh edges) and force signals (contact edges). Multiple message-passing rounds propagate deformation and stress information through the graph structure.

An ablation study compares two input configurations: force-only inputs (predicting gripper finger translation and deformation) and pre-specified translation inputs (predicting deformation only), with the latter simplifying the learning task.

## IV. RESULTS AND ANALYSIS

We evaluate the proposed model through a series of experiments of increasing complexity.

First, we train the model on individual objects to verify its ability to generalize to unseen grasping poses of known geometry. Table I demonstrates that providing translation input significantly reduces prediction error, as this simplification focuses learning on deformation dynamics without the need of predicting translation. This configuration also facilitates future sim-to-real transfer by aligning with real-world control paradigms.

Next, we scale to multi-object generalization using 10 objects (Fig. 3). The model is trained on 80% of grasping poses per object and tested on the remaining 20%. Figure 1 and Tab II shows predictions closely matching ground truth in the same range of single object training, demonstrating

the GNN's ability to generalize across objects with accuracy comparable. Additionally, similarly to single-object training, providing translation input reduces prediction error.

| Class | Trans | Mean Def MAE | Mean Stress MAE |
|-------|-------|--------------|-----------------|
| Average | True | **6.30e-05** | **360.3** |
|  | False | 2.69e-04 | 420.2 |

TABLE II: Multi-object training results averaged across all objects.

## V. LIMITATIONS AND FUTURE WORK

Two primary limitations are noted. First, the GNN's generalization to novel objects and tactile sensor configurations requires further evaluation. Expanding the model's scope to diverse geometries and sensor types would improve real-world applicability. Second, the current implementation neglects gravitational effects, which may affect force distribution predictions in dynamic scenarios.

## VI. CONCLUSIONS

This work demonstrates that GNNs provide an efficient alternative to FEM simulations for deformable tactile sensor prediction. Our framework accurately models gripper-object interactions, predicting both deformations and stress distributions while achieving $10^2$–$10^3\times$ speedup, enabling real-time performance. This fast, accurate model facilitates advanced grasp planning, tactile-based closed-loop control, and large-scale data generation for robotic manipulation.

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
