# OpenReview forum: "Real-Time Simulation of Deformable Tactile Sensors in Robotic Grasping using Graph Neural Networks"
_IEEE.org/IROS/2025/Workshop/Tactile_Sensing — IROS 2025 Workshop Tactile Sensing Poster_

### Official Review · Reviewer_CyFa · 2025-09-22
**Interesting proof-of-concept to shorten soft body simulation times**

**Rating:** 7
**Confidence:** 4

**Review:**

This is an interesting paper that uses GNNs to reduce the computational cost in performing soft body simulations. While the method is demonstrated on a set of objects and for a specific contact procedure, it is unclear what is the outlook to cover a broader set of scenarios.

It is unclear why GNNs make the approach so efficient -- is it because of sparse connections? Or what is the advantage compared to other learning architectures?

I highly recommend the authors to consider releasing the source code for this project, as tactile sensing simulation (especially FEM-based) is highly desirable in the community.

The paper exceeds the two-page length.

---

### Official Review · Reviewer_JMuy · 2025-09-24
**Promising method, but requires metrics and length reduction**

**Rating:** 7
**Confidence:** 4

**Review:**

This paper presents a GNN-based method for tactile sensor simulation, showing promising accuracy when compared to the ground truth. To strengthen the evaluation, I recommend quantifying the accuracy with common metrics (e.g., MAE, RMSE). Please also shorten the paper from its current three pages to the two-page workshop limit (excluding references).